# In Infants with Neuroblastoma Standard Therapy Only Partially Reverts the Fecal Microbiome Dysbiosis Present at Diagnosis

**DOI:** 10.3390/microorganisms13030691

**Published:** 2025-03-19

**Authors:** Maria Valeria Corrias, Eddi Di Marco, Carola Bonaretti, Margherita Squillario, Loredana Amoroso, Massimo Conte, Mirco Ponzoni, Roberto Biassoni

**Affiliations:** 1Laboratory of Experimental Therapies in Oncology, IRCCS Istituto Giannina Gaslini, Via Gerolamo Gaslini 5, 16147 Genova, Italy; mariavaleriacorrias@gaslini.org; 2Central Laboratory, IRCCS Istituto Giannina Gaslini, 00161 Genova, Italy; eddidimarco@gaslini.org; 3Molecular Diagnostics, IRCCS Istituto Giannina Gaslini, 00161 Genova, Italy; carolabonaretti@gmail.com (C.B.); robertobiassoni@gmail.com (R.B.); 4IRCCS Ospedale Policlinico San Martino, 00161 Genova, Italy; margherita.squillario@hsanmartino.it; 5Oncology Unit, IRCCS Istituto Giannina Gaslini, 00161 Genova, Italy; l.amoroso@policlinicoumberto1.it (L.A.); massimoconte@gaslini.org (M.C.)

**Keywords:** fecal microbiome, infants, neuroblastoma, chemotherapy, 16S sequencing

## Abstract

The fecal microbiomes of 15 infants with neuroblastoma (NB) at the onset of the disease and after standard-of-care therapy have been prospectively analyzed compared to those of age-matched healthy infants. By applying several algorithms to 16S sequencing, we found that the fecal microbiomes of infants with NB at onset were abundant in *Pseudomonadota*, including different descendants of *Gammaproteobacteria.* After completing therapy, their abundance decreased to a level like that observed in healthy infants. In contrast, the *Bacillota* that showed at the onset an abundance like that of healthy infants doubled their amount after treatment. In infants with NB, the beta diversity of the fecal microbiomes was significantly reduced compared to healthy infants and patients at the end of treatment. The Random Forest algorithm and the Reingold–Tilford heat tree showed that *Enterobacteriaceae* had a higher abundance at the onset, which declined after therapy. Picrust2 inferred pathway analysis indicated that the drug treatment was associated with a reduction in the polyamine pathway, highly represented in samples of NB at the onset. In conclusion, the dysbiosis observed in infants with NB at onset changed following standard-of-care treatment. Still, the composition at the end of treatment did not completely resemble that of healthy infants.

## 1. Introduction

Neuroblastoma (NB) is the most common extracranial solid tumor in children, accounting for approximately 8% of all pediatric cancer cases and 15% of related deaths [1]. It originates from neural crest progenitor cells, and the primary tumor displays specific genetic and biological characteristics crucial in determining its clinical outcome [2]. Age at diagnosis is one of the most significant prognostic factors [3]. Specifically, infants diagnosed before 18 months have an overall survival rate greater than 95%, following a standard treatment approach that includes surgery and mild chemotherapy [4]. In contrast, children diagnosed after 18 months have an overall survival rate ranging from 75% to 50%, depending on the presence of *MYCN* amplification or metastatic disease, respectively [4]. Neuroblastoma (NB) is generally more common at very young ages, with an average diagnosis age of 20 months [1,2,3], which has led to the hypothesis that environmental factors may play a role in its etiology. Since in recent years, the environmental factor found to be non-randomly associated with multiple types of cancer in the adult population has been the gut microbiome [5], we recently analyzed the fecal microbiome composition of children with NB at disease onset, comparing it to a group of healthy controls matched for key covariates, such as age, sex, stool consistency, mode of delivery, and feeding method [6]. The study revealed significant differences in the gut microbiome of children with NB compared to healthy children, including their siblings. The observed variations were not linked to changes in maternal seeding; rather, they appeared to stem directly from the disease itself [6].

Given that the gut microbiome composition in infants (aged 0–24 months) differs from that in older children (aged > 24 months) [7] and that survival rates are significantly higher in infants compared to older children [4], here, we prospectively analyzed the fecal microbiome of infants with NB at disease onset and after standard care treatment with carboplatin/VP16. The goal was to assess whether the initial dysbiosis was restored to a healthy gut microbiome following treatment.

## 2. Methods

### 2.1. Study Design

Between September 2020 and September 2022, the fecal samples of 15 infants aged < 24 months (10 ± 7 months old) diagnosed with NB at the IRCCS Istituto Giannina Gaslini were collected before starting any therapy, following written consent from the legal guardians. During the same period, fecal samples were collected from 17 age-matched healthy infants (HC, 14 ± 6 months old) and used as a reference cohort. Fecal samples were also collected from infants with neuroblastoma (NB) after two cycles of chemotherapy with carboplatin and VP16, which is considered standard care (Table 1). Five infants participated in this study. The first sample was taken two months into therapy, representing half the treatment duration. The second sample was collected after four cycles of chemotherapy at the end of four months of drug administration.

The study received approval from the Regione Liguria Ethics Committee (N° 006/2019), and all experiments were conducted in accordance with the Helsinki Declaration.

Sample collection was performed using OMNIgene-Gut OMR-200 devices, which contain a stabilization solution for the microbial DNA in fecal content (DNAgenotek, Ottawa, ON, Canada). The biological materials were stored at −20 °C until microbial DNA extraction.

To extract DNA, 200 microliters of fecal suspension from the OMR-200 devices were vigorously pipetted into 1 mL of ASL Stool lysis buffer (Qiagen, Hilden, Germany). The samples were then processed using the MagDEA DNA 200GC extraction kit and the PSS Magtration System 12GC automated platform, following the manufacturer’s instructions (Precision System Science PSS Co., Ltd., Matsudo, Japan).

The extracted DNA was eluted into 100 microliters of 10 mM TE buffer, and its quality and quantity were evaluated using spectrophotometric and Qubit fluorimetric quantitation assays, respectively (ThermoFisher Scientific, Waltham, MA, USA). To perform each 16S amplification reaction, 3 ng of DNA was used with the Ion 16S Metagenomics Kit™ (ThermoFisher Scientific). This kit allows the analysis of 7 out of 9 informative 16S polymorphic regions (V2, V4, V8, V3, V6–7, V9) through PCR amplification. After amplification, library preparation was performed using the IonPlus-Library kit for the AB library builder instrument (ThermoFisher Scientific). Different bar-coded libraries were created and automatically loaded onto an Ion 520 chip using the Ion Chef System and sequenced on the GeneStudio S5 system (ThermoFisher Scientific). Data analysis was conducted through the Ion Reporter™ suite software (v 5.18) utilizing the curated Greengenes (v13.5) and premium curated MicroSEQ ID 16S rRNA reference library (v2013.1) databases with standard parameters. Taxonomy followed the LPSN web interface (https://www.bacterio.net [8], accessed on 5 March 2025).

The mean 16S mapped reads per sample, obtained from all the fecal microbiome specimens was 207,285+/−55,002. These reads identified 1534+/−388 operational taxonomic units (OTUs) in the 26 NB samples and 1416+/−395 OTUs in the 17 HC specimens (Table 1).

### 2.2. Alpha and Beta Diversity Analysis

Alpha and beta diversity analysis are important techniques in ecology. Various alpha diversity indices are used to estimate community richness, such as the Chao1 index [9], or to assess both richness and evenness, as seen with the Shannon or Simpson indexes [10,11]. For beta diversity analysis, the Bray–Curtis index (or Bray–Curtis dissimilarity) is a statistical method that quantifies the differences in microbial composition between two samples [12]. PERMANOVA (Permutational Multivariate Analysis of Variance) is another statistical method commonly used to analyze multivariate data. When paired with the Bray–Curtis dissimilarity metric, PERMANOVA evaluates whether significant differences exist between groups based on community composition, such as species abundance in various habitats.

### 2.3. Bioinformatics and Statistics

To analyze the microbiome composition and its functions, we employed the MicrobiomeAnalyst suite [13,14,15]. We filtered out the low-abundance and low-variance OTUs, based on their prevalence in 20% of the samples and, with the interquartile range set at 10%, respectively. For relative abundance comparisons, we used multiple data filtering and normalization techniques coupled with different algorithms of analysis, like metagenomeSeq (based on zero-inflated Gaussian fit statistical models), EdgeR, DESeq2, linear discriminant analysis (LDA), and effect size analysis (LEfSe) [16,17,18,19]. To correct for multiple hypotheses, all *p* values have been adjusted using the Benjamini and Hochberg false discovery rate (FDR < 0.05) unless otherwise specified [20].

To analyze changes in the microbiome composition of patients before and after therapy, additional methods were employed, including the Reingold–Tilford hierarchical heat tree analysis, Sparse Correlations for Compositional Data (SparCC), and Random Forest machine learning.

The Reingold–Tilford hierarchical tree analysis compares the microbial communities pairwise. This analysis measures quantity using median abundance and identifies statistically significant taxonomic variations among microbial communities through the non-parametric Wilcoxon Rank Sum test, adjusted for multiple comparisons using FDR correction [21]. Sparse Correlations for Compositional Data (SparCC) is a computational method to analyze microbiome data, such as 16S rRNA sequencing outputs, where the observed values represent relative abundances rather than absolute counts [22]. Traditional correlation methods like Pearson or Spearman can be unreliable for such data due to the inherent constraints of compositional datasets (e.g., the sum of all components is constant). SparCC overcomes these limitations by modeling correlations while accounting for the compositional nature of the data. SparCC utilizes log-ratio transformations to address proportionality, reducing biases associated with conventional correlation metrics. It assumes that only a limited subset of taxa interacts directly, resulting in a sparse correlation network. This assumption simplifies analysis and enhances interpretability. Using sparse inverse covariance estimation, SparCC identifies direct interactions between taxa while accounting for indirect effects mediated through other taxa. By estimating pairwise correlations, it constructs interaction networks among microbial taxa, revealing ecological relationships such as cooperation or competition. Furthermore, SparCC can identify keystone taxa-species essential for maintaining the stability and structure of microbial communities. Widely used in microbiome research, SparCC aids in uncovering microbial co-occurrence patterns, identifying taxonomic interactions, and understanding ecosystem dynamics, particularly in the context of health, disease, and environmental microbiomes. By effectively addressing the challenges of compositional data, SparCC is a powerful tool for interpreting complex relationships within microbial ecosystems. Moreover, the microbial dysbiosis index (MD-index) was calculated as the natural logarithmic ratio between the sum of the abundance of all taxa that increased their value and the sum of the ones that decreased [23]. The MD-index measures the microbial community’s imbalance or deviation from a healthy or normal state. It is computed by comparing the relative abundance of different microbial taxa to a reference or control population. A value close to zero indicates a normal or healthy microbial population, whereas a value above or below zero indicates a degree of dysbiosis or abnormal microbial composition.

A Random Forest is an ensemble machine-learning algorithm primarily used for classification and regression tasks. It works by constructing multiple decision trees, each built from a random sample (with replacement) of the training data, creating a more robust model. During training, each tree is formed from a random subset of the data using techniques like bagging (bootstrap aggregating). For classification, the final output is determined by the mode of the individual trees’ predictions, while for regression, it is the mean of those predictions. By combining many trees, Random Forest helps reduce the risk of overfitting, which happens with individual decision trees.

Additionally, it provides feature importance estimates, aiding in understanding the contribution of each feature to the model’s predictions. Mean decrease accuracy is a metric used to assess the importance of each feature in a Random Forest model. It quantifies how much the model’s accuracy decreases when the values of a specific feature are randomly permuted while keeping the other features unchanged. Features that significantly impact the model’s performance will cause a higher decrease in accuracy when permuted, indicating higher importance.

The weighted correlation network analysis (WGCNA) examined the relationship between taxa and clinical parameters. WGCNA identified a cluster of taxa that showed a strong correlation with each other across different samples or conditions [24]. This analysis assists in identifying key taxa linked to specific disease characteristics. Using this method, taxa that displayed similar behavior were grouped into modules or clusters, and their association strength was measured using a weighted correlation network. The evaluation included the following criteria: Sex: M = 1, F = 0; Staging: L1 = 1, L2 = 2, M and Ms = 3; Age: continuous variable; Delivery: Eutocic = 1, C-section = 0; Feeding: Breast = 1, Formula = 0; Bristol score 1 and 2 (severe or mild constipation) = 1; score 3 and 4 (normal behavior) = 2; score 5, 6, and 7 (mild and severe diarrhea) = 3.

To obtain the table of KEGG Orthology (KO) terms [25], the functional content of the metagenome was predicted using PICRUSt2 [26]. These data were then analyzed using the Shotgun-data-profiling module of MicrobiomeAnalyst to identify the pathways most significantly associated with infants with NB at the onset or after completion of therapy. The results were subsequently integrated with those obtained from the Shotgun-data-profiling module of MicrobiomeAnalyst.

## 3. Results

### 3.1. Alpha and Beta Diversity at Onset and During Treatment

The Chao1 index of the fecal microbiome in infants with NB at onset was slightly lower than that found in healthy infants. The microbiome richness increased after therapy, but the difference was not statistically significant. The Bray–Curtis dissimilarity index shows that the microbiomes of infants with NB were significantly different from those of healthy infants both at onset (Appendix A, *p* = 0.006) and after two cycles of therapy (Appendix A, *p* = 0.021) whereas, after four cycles of drug treatment, the difference was not significant (Appendix A). However, the microbiome of infants with NB after two or four cycles of therapy was not significantly different from those of NB at onset (Appendix A, respectively), confirming that the pharmacological treatment did not fully restore the fecal microbiome composition found in healthy infants, with the difference still being significant (*p* = 0.002, Appendix A).

### 3.2. Microbiome Compositional Analysis of Healthy Infants and Infants with NB

Infants with neuroblastoma (NB) exhibit a distinct composition of fecal bacteria compared to healthy infants. All four algorithms used for the analysis assessed that the fecal microbiome of infants with neuroblastoma (NB) at onset had a higher abundance of *Pseudomonadota* compared to age-matched healthy infants (Appendix A). Indeed, as shown in Figure 1A, the fecal microbiome of infants with NB was mainly composed of the *Pseudomonadota* phylum (median of abundance 39.3% vs. 12.1% in healthy controls). Interestingly, *Pseudomonadota* were sensitive to the drug treatment, as their abundance at the end of therapy was only 18.9%, comparable to the quantity found in healthy infants. At the onset, the abundance of *Bacillota* in infants with neuroblastoma (NB) was like that of the control group. Unlike *Pseudomonadota*, the abundance of *Bacillota* doubled by the end of the treatment (Figure 1B). Differently, the abundance of *Bacteroidota* (Figure 1C) was significantly lower in infants with neuroblastoma (NB), both at the beginning of the study (1.7%) and after therapy (0.1%), compared to healthy infants (41.4%). Finally, the abundance of *Actinomycetota* (Figure 1D) was similar in infants with NB at the onset (9.2%) and in healthy infants (10.2%). However, there was a significant decrease in the abundance of *Actinomycetota* in infants with NB by the end of therapy (1.2%). To summarize, the primary type of bacteria found in the feces of infants with NB is *Pseudomonadota*, which is more than three times more abundant than in the control group. In contrast, levels of *Bacteroidota* are significantly lower in these infants, while *Bacillota* and *Actinomycetota* exhibit similar abundances to those found in healthy controls.

Analyses of genera abundance in the fecal microbiome of infants with neuroblastoma (NB) at onset, compared to healthy infants, indicated a higher abundance of 13 genera, 8 belonging to the *Pseudomonadota* (Figure 2A). In detail, seven of them (*Enterobacter*, *Escherichia*, *Escherichia/Shigella*, *Klebsiella*, *Leclercia*, *Salmonella*, and *Trabulsiella*) belong to the class of *Gammaproteobacteria*. Conversely, the genera that showed a higher abundance in healthy infants than in infants with NB, apart from the *Prevotella,* belonged to the *Bacillota* phylum (Figure 2A, Appendix A). During treatment (after two cycles of chemotherapy), the number of genera significantly different between patients and controls decreased (Figure 2B, Appendix A). After completing four cycles of therapy, the abundance of *Lactobacillales*, particularly from the *Enterococcus* and *Streptococcus* genera, increased compared to the fecal microbiome of healthy infants (Figure 2C, Appendix A). After treatment, the *Enterobacter* genus was found to be more abundant than at the onset, while there was a significant decrease in the presence of taxa such as *Prevotellaceae*, *Campylobacteraceae*, and the species *Turicibacter sanguinis* (see Figure 2D and Appendix A). At the end of treatment, the abundance of several bacterial families from the *Bacillota* phylum, including *Clostridiaceae*, *Lachnospiraceae*, and *Acidaminococcaceae*, significantly increased compared to the halfway point of the therapy (Appendix A).

### 3.3. Supervised Analysis with Random Forest Learning Algorithm

The Random Forest machine learning algorithm, applied to the comparison of fecal microbiomes of infants with NB at onset and age-matched healthy infants agreed with the data obtained from differential abundance algorithms 10 times out of 15, indicating that a microorganism belonging to a similar taxonomy was identified in 70% of cases to the correct taxon (Appendix A). The fecal microbiome composition in infants with neuroblastoma changed significantly after drug treatment, with *Enterobacteriaceae*, which showed a substantial decrease in their abundance after just two cycles of drug treatment (Appendix A). Noticeable differences were no longer statistically significant after four cycles of treatment (end of therapy) (Appendix A).

### 3.4. Heat Tree Analysis

In the Reingold–Tilford heat tree, the hierarchical structure of taxonomic classifications evaluated the variations in microbial communities. As shown in Figure 3A, *Lactobacillaceae*, *Lachnospiraceae*, *Eubacteriaceae*, *Bacteroides*, and *Pasteurellaceae* were more abundant in the fecal microbiome of healthy infants than in infants with NB at the onset. Conversely, *Enterobacteriaceae*, *Enterococcaceae*, and their descendants were more prevalent in the fecal microbiome of infants with NB than in healthy infants. By the end of therapy, the predominant feature of the fecal microbiome in NB was a decreased abundance of *Actinomycetota* and their descendants (Figure 3B).

### 3.5. Sparse Correlations for Compositional Data (SparCC) Analysis

In infants with NB at onset, different genera belonging to the *Gammaproteobacteria*, like *Enterobacter*, *Citrobacter, Klebsiella, Leclercia, Escherichia, Escherichia-Shigella, Morganella, Trabulsiella, Klebsiella, Salmonella, Serratia*, and *Yokenella* were more abundant than in healthy infants (Figure 4A). In contrast, other *Gammaproteobacteria*, such as *Mannheimia* and *Haemophilus*, were more abundant in healthy infants. Similarly, many descendants of the phylum *Bacillota*, like *Dialister, Eubacterium, Roseburia, Faecalibacterium*, and some *Bacteroidota*, like *Alistipes, Prevotella*, and *Bacteroides*, showed higher abundance in healthy infants. Intriguingly, as shown in Figure 4B, after completion of therapy, for various genera belonging to *Pseudomonadota*, the orange surface area associated with NB pathology decreased in size (*Citrobacter, Cronobacter, Escherichia, Haemophilus, Klebsiella*, and *Trabulsiella*) or lost statistical significance (*Escherichia-Shigella, Gemminger, Leclercia, Salmonella*, and *Yokenella*). On the opposite end, different *Bacillota* descendants increased their orange surface after therapy (*Blautia, Dialister, Eubacterium, Faecalibacterium, Roseburia, Ruminococcus, Streptococcus*, and *Subdoligranulum* in order of magnitude), confirming a different behavior of the two phyla following standard of care therapy.

**Figure 3 microorganisms-13-00691-f003:**
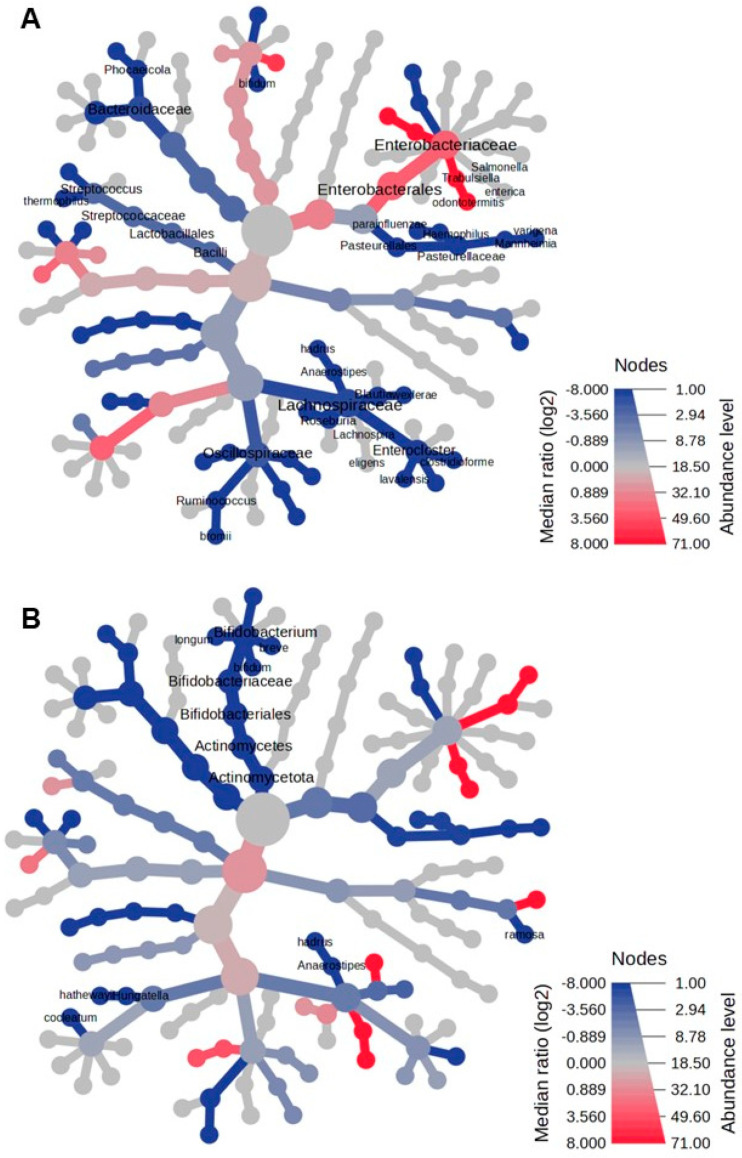
The heat tree comparison illustrates the differences in taxonomic abundance between the fecal microbiomes of two sample groups, analyzed using a non-parametric Wilcoxon Rank Sum test. The comparisons are made between the following: (**A**) patients with neuroblastoma (NB) at the onset of the disease versus healthy controls, and (**B**) patients with NB after completing therapy versus healthy controls. Shades of blue indicate a negative median ratio of abundance levels, represented as log2 fold changes (log2FC), while shades of red signify a positive median ratio log2FC in the comparison between groups.

### 3.6. Microbial Dysbiosis Index

The microbial dysbiosis index was calculated by comparing the microbiome of infants with neuroblastoma with the group of healthy infants. The index values were as follows: 1.6048 for NB samples at onset, 0.9 for NB samples after two cycles of therapy, and −1.05 after four cycles of drugs.

### 3.7. Weighted Correlation Network Analysis

Using the WGCNA, we then assessed whether groups of organisms were related to specific characteristics of the infants with NB, such as sex, stage, age, type of delivery, feeding method, and Bristol score. Our findings, outlined in Table 2 and depicted in Appendix A, revealed positive correlations between groups of organisms and the age of patients. Additionally, we observed a negative correlation between the abundance of *Clostridioides difficile*, *Enterobacter* sp., *Kluyvera georgiana*, and the Bristol score. These data indicated that a paucity of these species correlated with mild and/or severe diarrhea.

### 3.8. Differentially Expressed Inferred Pathways by the Fecal Microbiome of Infants with NB

The abundance of inferred pathways obtained with Picrust2 in the microbiome of infants with NB at onset and after therapy completion was compared using both EdgeR and DESeq2 statistical methods. EdgeR analysis indicated that 1074 keggs orthology (ko) pathways were significantly different with an FDR < 0.05, and 845 keggs ortholog with an FDR < 0.01 (Appendix A). By using the DESeq2 algorithm only 23 of the 1074 (FDR < 0.05) and only 9 of the 845 (FDR < 0.01) from the EdgeR pathways were confirmed (Table 3). It is interesting to note that 5 of the 23 pathways defined by both algorithms decreased in abundance after completion of therapy and that 3 of them were metabolic pathways related to polyamines (putrescine and spermidine).

## 4. Discussion

The richness of the fecal microbiome in infants with neuroblastoma at onset was marginally lower than that of healthy infants. Although there was an increase in richness after therapy, the change was not statistically significant. Similarly, diversity (evenness) improved after treatment but remained below the levels observed in healthy infants, indicating an incomplete restoration of a healthy microbiome. Microbial composition in infants with NB was significantly different from healthy infants at both onset and after two cycles of therapy. However, this distinction became insignificant after four cycles, suggesting a convergence toward healthy microbiomes, albeit incomplete.

In infants, the fecal microbiome is a dynamic community that evolves significantly during early life and reaches an adult-like composition by 24–36 months of age [7,27]. Its composition is influenced by birth modality, environmental exposure, feeding practices, and dietary transitions during weaning [28]. Our findings are consistent with a previous study involving older children with NB [6], extending its conclusions to infants.

This study employed 16S rRNA sequencing to analyze the fecal microbiome of infants less than 24 months old. Despite inherent variability, robust analytical approaches revealed notable features in the microbiome of infants with NB. At disease onset, *Pseudomonadota* were significantly overrepresented, comprising 39.3% of the microbiome compared to 12.1% in healthy infants. Post-therapy, this proportion decreased to 18.9%, closer to healthy levels. Within *Pseudomonadota*, genera such as *Enterobacter*, *Escherichia*, and *Klebsiella*, known for containing pro-inflammatory and pathogenic species associated with gut dysbiosis [27,28,29,30], were highly abundant at the onset. Although the reduction in *Pseudomonadota* abundance post-therapy is a positive effect, it was insufficient to restore the normal composition of the fecal microbiome or eubiosis.

After four treatment cycles, notable shifts were observed in microbiome composition at the phyla level. Indeed, *Bacillota* showed a doubling in abundance, *Actinomycetota* declined to a quarter of their original abundance, and the *Bacteriodota* (e.g., *Prevotella*) and the *Bacillota* family of *Clostridiaceae* remained depleted, only partially recovering post-therapy.

The incomplete restoration of the microbiome indicates a persistent imbalance. Notably, families such as *Enterococcaceae* and *Streptococcaceae*, which are highly abundant at disease onset, showed resilience and did not decrease after treatment. The adaptability of enterococci, capable of surviving harsh conditions, may explain their persistence [31].

Advanced analytical methods confirmed these patterns, with the most striking taxonomic changes occurring in *Enterobacteriaceae* during therapy. Age-related correlations and associations between specific taxa (e.g., *Clostridioides difficile*, and species belonging to *Enterobacter* sp.) and clinical symptoms such as diarrhea (assessed via the Bristol stool score) were identified.

Inferred pathway analysis revealed 23 consistently altered pathways. At the onset, metabolic pathways related to polyamines (e.g., putrescine, spermidine) were highly abundant and decreased significantly after therapy, in full accordance with previous analyses of the metabolome in human [32] and murine NB [33]. Polyamines like putrescine and spermidine are associated with several important cellular functions and might have pleiotropic effects [34]. Polyamines are essential for cellular functions and have been implicated in creating a pro-tumorigenic microenvironment in cancer patients [35,36]. Thus, their reduction following therapy may contribute to its clinical efficacy.

Although our study has been performed in a small cohort, the use of different tools to analyze the data showing consistent results in a population with the higher variability in the microbiome composition, such as infants less than 2 years old at diagnosis [7,28], supports the fact that neuroblastoma affects the gut microbioma with long-lasting effects that may have relevant health implications. Therefore, future research should explore whether interventions, like the addition of probiotics, prebiotics, and nutritionally active substances [37,38] to promote a full microbiome eubiosis at the end of therapy, by restoring beneficial taxa, such as *Bacteroidota* and *Actinomycetota,* may improve outcomes.

In conclusion, standard of care treatment in infants with NB significantly modulated the altered fecal microbiome present at diagnosis, but a full restoration to a healthy state was not achieved. If future studies demonstrate that the gut microbiome dysbiosis also persists in children older than 24 months at diagnosis and following a more intense therapeutic regimen, functional studies on therapy-impacted metabolic pathways may provide insights into their role in disease progression, opening new avenues for treatment strategies of high-risk NB.

## Figures and Tables

**Figure 1 microorganisms-13-00691-f001:**
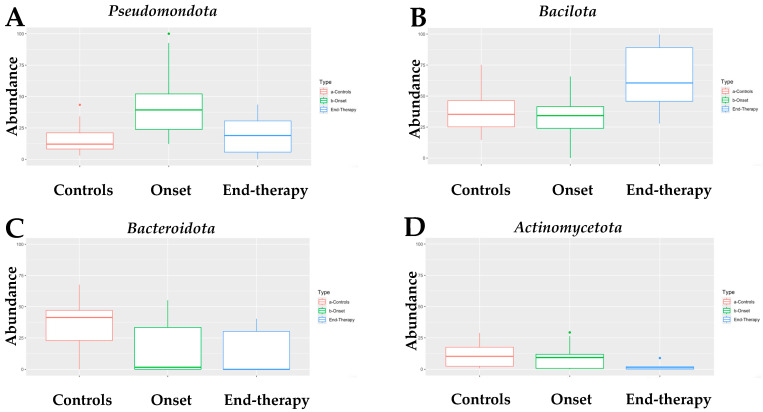
Box-plot of the abundance of (**A**) *Pseudomonadota*, (**B**) *Bacillota*, (**C**) *Bacteroidota*, and (**D**) *Actinomycetota* in the fecal microbiome of NB patients at onset (N = 15) and at the end of treatment (N = 6), as compared with that of healthy children (controls, N = 17). Dots in panel (**A**,**D**) of Figure 1 indicate statistical significance (*p* < 0.05) when comparing the abundance of *Pseudomonadota* in samples at the onset versus the controls (**A**) and of *Acinomycetota* in specimens at the onset versus the one at the end of therapy (**D**).

**Figure 2 microorganisms-13-00691-f002:**
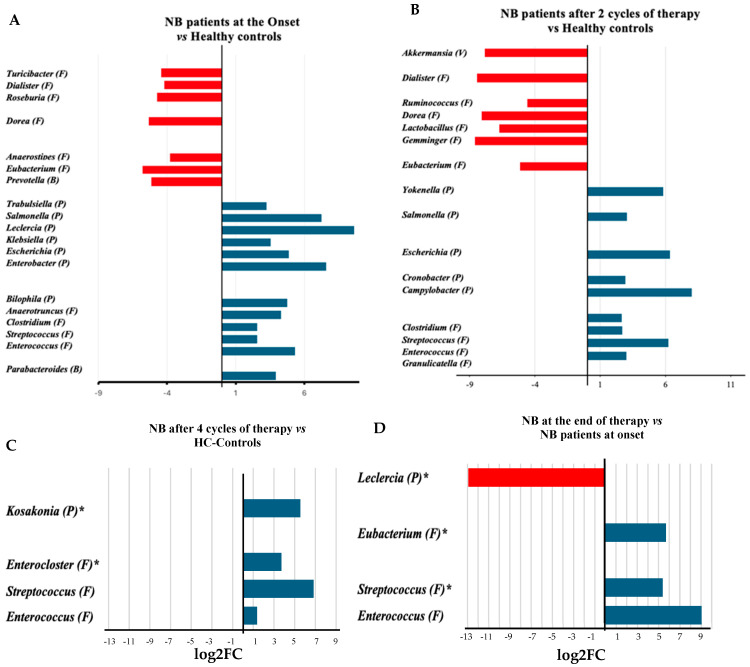
Log2 fold change (log2FC) in the abundance of microbial genera between the microbiomes of infants with neuroblastoma at onset [15] (**A**), or after 2 cycles of treatment [5] (**B**) and at the end of therapy [6] (4 cycles, (**C**)) in comparison with the microbiomes of healthy controls (HC) [17] or between samples after the four treatment cycles [6] and samples at onset [15] (**D**). The number in squared parenthesis indicates the number of subjects representing each group in the analysis. The log2FC was calculated using the EdgeR algorithm. The false discovery rate (FDR) indicates the *p*-value after adjustment for multiple comparisons. The FDR was <0.001, except for the genera shown with the asterisk (*, FDR < 0.05). The phylum classification is shown in round parentheses as B: *Bacteroidota*, F: *Bacillota*, P: *Pseudomonadota*, and V: *Verrucomicrobiota*.

**Figure 4 microorganisms-13-00691-f004:**
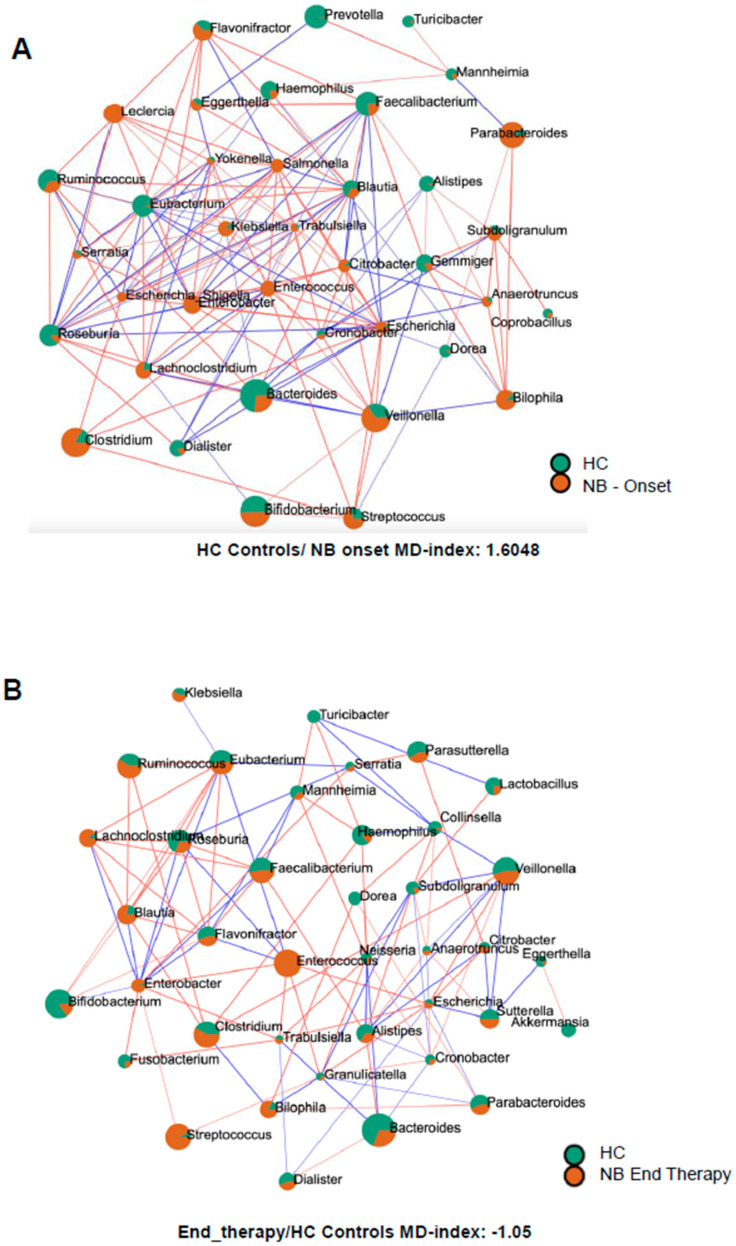
SparCC correlation analysis between the microbiomes of healthy infants (HC) and of infants with NB at onset (**A**) or after 4 cycles of therapy (**B**). Circles represent genera, the color (orange = NB, green = HC) reflects the abundance in each group and the size depends on the number of connections. The line color, red or blue, represents a positive or negative correlation, respectively.

**Table 1 microorganisms-13-00691-t001:** Neuroblastoma patients at onset and after therapy.

(A)									
Sample	Sex	Phase	Stage	Age (Months)	Mode of Delivery	Mode of Feeding	Bristol Score	READS	OTUs
P1	F	onset	MS	6.6	eutocic	breast	7	240,098	1879
P2	M	onset	L1	10.9	eutocic	formula	4	232,017	1491
P3	M	onset	L1	10.9	c-section	formula	2	224,975	1578
P5	F	onset	L2	8.4	eutocic	formula	1	349,389	2567
P7	F	onset	L2	22.0	eutocic	formula	2	184,108	1435
P8	F	onset	M	19.8	c-section	breast	6	228,071	1623
P10	M	onset	MS	4.8	eutocic	breast	6	197,682	1311
P11	M	onset	L1	2.6	c-section	breast	3	313,173	2306
P12	F	onset	MS	4.9	c-section	formula	7	186,662	1445
P15	M	onset	M	22.1	eutocic	formula	3	228,290	1488
P19	F	onset	M	5.8	c-section	breast	6	130,251	842
P21	F	onset	L2	6.0	eutocic	breast	6	240,067	1891
P22	M	onset	M	8.6	eutocic	breast	6	182,989	1364
P24	M	onset	L1	5.0	eutocic	breast	6	128,618	971
P25	M	onset	L2	6.5	eutocic	formula	6	208,146	1419
(**B**)									
**Sample**	**Sex**	**Phase**	**Stage**	**Age (Months)**	**Drug Therapy**		**Bristol Score**	**READS**	**OTUs**
P1 B	F	CP/VP16	MS	9.6	2 cycles—half treatment		7	251,064	1660
P7 A	F	CP/VP16	L2	24	2 cycles—half treatment		6	229,871	1722
P21 A	F	CP/VP16	L2	8.4	2 cycles—half treatment		6	197,013	1714
P22 A	M	CP/VP16	M	9.6	2 cycles—half treatment		6	182,989	1683
P24 A	M	CP/VP16	L1	8.4	2 cycles—half treatment		6	130,474	1355
P1 C	F	CP/VP16	MS	12	4 cycles—end of treatment		6	145,892	970
P5 A	F	CP/VP16	L2	21.6	4 cycles—end of treatment		4	109,008	944
P7 B	F	CP/VP16	L	31.2	4 cycles—end of treatment		5	217,098	1438
P12 A	F	CP/VP16	MS	8.4	4 cycles—end of treatment		6	239,966	1654
P22 B	M	CP/VP16	M	13.2	4 cycles—end of treatment		6	221,960	1619
P24 B	M	CP/VP16	M	20.4	4 cycles—end of treatment		7	152,753	1253

Some of the patients, after diagnosis, were grouped and studied during treatment.

**Table 2 microorganisms-13-00691-t002:** WGCNA on fecal microbiomes of NB patients at onset and the correlation with clinical variables.

		Age		Bristol Score	
	*Family*	Pearson Corr. Coeff.	*p*-Value	Pearson Corr. Coeff.	*p*-Value
**Cluster 7**	*Christensenellaceae*, *Desulfohalobiaceae*, *Desulfovibrionaceae*, *Erysipelotrichaceae*, *Eubacteriaceae*, *Oscillospiraceae*, *Paenibacillaceae*, *Peptococcaceae*, *Porphyromonadaceae*, *Rikenellaceae*, *Verrucomicrobiaceae*	0.59	0.020		
**Cluster 8**	*Bacteroidaceae*, *Campylobacteraceae*, *Dethiosulfovibrionaceae*, *Hyphomicrobiaceae*, *Lachnospiraceae*, *Micrococcaceae*, *Oscillospiraceae*, *Prevotellaceae*, *Synergistaceae*	0.54	0.04		
	** *genus* **				
**Cluster 1**	*Akkermansia*, *Alistipes*, *Anaerofustis*, *Anaerotruncus*, *Barnesiella*, *Bilophila*, *Desulfovibrio*, *Flavonifractor*, *Hydrogenoanaerobacterium*, *Moryella*, *Oscillibacter*, *Parabacteroides*, *Subdoligranulum*	0.58	0.02		
**Cluster 7**	*Abiotrophia*, *Butyricicoccus*, *Eggerthella*, *Enterococcus*, *Howardella*, *Megamonas*, *Parasutterella*, *Pseudomonas*	0.54	0.04		
	** *species* **				
**Cluster 6**	*Clostridioides_difficile*, *Kluyvera_georgiana*			−0.53	0.04
**Cluster 12**	*Abiotrophia_defectiva*, *Bacteroides_ovatus*, *Bifidobacterium_bifidum*, *Bifidobacterium_breve*, *Butyricicoccus_pullicaecorum*, *Clostridium_innocuum*, *Eggerthella_lenta*, *Enterocloster_citroniae*, *Enterococcus_avium*, *Enterococcus_faecium*, *Enterococcus_lemanii*, *Megamonas_funiformis*, *Megamonas_rupellensis*, *Pseudoflavonifractor_capillosus*, *Pseudomonas_aeruginosa*, *Pseudomonas_indica*	0.52	0.04		
**Cluster 17**	*Akkermansia_muciniphila*, *Anaerofustis_stercorihominis*, *Anaerotruncus_colihominis*, *Barnesiella_intestinihominis*, *Blautia_hansenii*, *Eggerthella_sinensis*, *Eubacterium_callanderi*, *Flavonifractor_plautii*, *Moryella_indoligenes*, *Subdoligranulum_variabile*	0.55	0.04		

Weighted Gene Co-expression Network Analysis (WGCNA) is used to find clusters of highly correlated features in large datasets. When applied to fecal microbiomes and clinical variables, WGCNA can identify clusters of microbial taxa that share variation patterns with clinical traits across samples. For each analysis between microbial clusters and clinical variables, the Pearson correlation coefficient and its statistical *p*-values are shown. Grouping values used in the analysis: **Sex:** M = 1, F = 0; **Stage:** L1 = 1, L2 = 2, M and Ms = 3; **Age:** continuous variable; **Delivery:** Eutocic = 1, C-section = 0; **Feeding:** Breast = 1, Formula = 0; **Bristol score:** Type 1 and 2 (severe or mild constipation) = 1; Type 3 and 4 (normal) = 2; Type 5, 6 and 7 (mild and severe diarrhea) = 3.

**Table 3 microorganisms-13-00691-t003:** Statistically relevant inferred pathways were obtained by analyzing fecal microbiomes of NB patients after completing therapy and comparing them to samples of NB at onset using Picrust2, DESeq2, and EdgeR algorithms.

ko	DESeq2				EdgeR				Pathways	Orthology
	log2FC	lfcSE	*p*-Values	FDR	log2FC	logCPM	*p*-Values	FDR		
K03190	32.498	0.62533	2.03 × 10^−3^	0.00079121	3.2919	6.1374	3.3685 × 10^−11^	2.0242 × 10^−9^		ureD, ureH; urease accessory protein
K01428	31.685	0.62601	4.16 × 10^−3^	0.0008125	3.2364	6.1378	6.5176 × 10^−11^	3.1912 × 10^−9^	ko:K01428 ureC; urease subunit alpha [EC:3.5.1.5]	
K01071	40.126	0.8589	2.98 × 10^−2^	0.0038861	5.4371	8.745	1.2689 × 10^−12^	2.9721 × 10^−10^	ko:K01071 MCH; medium-chain acyl-[acyl-carrier-protein] hydrolase [EC:3.1.2.21]	
K07146	26.275	0.5798	5.85 × 10^−2^	0.0057137	4.0782	8.8766	5.3469 × 10^−12^	6.5049 × 10^−10^		K07146; UPF0176 protein
K00563	19.082	0.42737	8.00 × 10^−2^	0.0061473	3.4302	9.0049	2.6382 × 10^−12^	4.9071 × 10^−10^		rlmA1; 23S rRNA (guanine745-N1)-methyltransferase [EC:2.1.1.187]
K01478	21.936	0.49523	9.44 × 10^−2^	0.0061473	4.6804	8.3424	4.0374 × 10^−15^	3.9425 × 10^−12^	ko:K01478 arcA; arginine deiminase [EC:3.5.3.6]	
K07586	34.572	0.78697	1.12 × 10^−1^	0.0062367	4.9499	8.7905	7.5016 × 10^−12^	8.1392 × 10^−10^		ygaC; uncharacterized protein
K00383	23.032	0.52916	1.35 × 10^−1^	0.0065712	4.246	9.607	3.161 × 10^−13^	1.5434 × 10^−10^	ko:K00383 GSR; glutathione reductase (NADPH) [EC:1.8.1.7]	mprF, fmtC; phosphatidylglycerol lysyltransferase [EC:2.3.2.3]
K02052	−16.149	0.37679	1.82 × 10^−1^	0.0078962	−2.3178	8.0724	0.00012877	0.001267	ko:K02052 ABC.SP.A; putative spermidine/putrescine transport system ATP-binding protein	
K01674	30.408	0.7332	3.36 × 10^−1^	0.013141	5.3147	8.3658	1.6729 × 10^−14^	1.089 × 10^−11^	ko:K01674 cah; carbonic anhydrase [EC:4.2.1.1]	
K04784	−44.021	10.957	5.88 × 10^−1^	0.019938	−5.266	8.7619	0.00014225	0.0013821	ko:K04784 irp2; yersinia b-actin nonribosomal peptide synthetase	
K00244	14.086	0.35146	6.13 × 10^−1^	0.019938	1.7709	9.5778	0.00092	0.0052846	ko:K00244 frdA; succinate dehydrogenase flavoprotein subunit [EC:1.3.5.1]	
K15051	27.683	0.6988	7.45 × 10^−1^	0.022379	3.159	9.3922	1.0309 × 10^−5^	0.00015025		endA; DNA-entry nuclease
K14260	1.606	0.40857	8.46 × 10^−1^	0.023616	2.9388	9.2072	9.3617 × 10^−10^	3.4826 × 10^−8^	ko:K14260 alaA; alanine-synthesizing transaminase [EC:2.6.1.66 2.6.1.2]	
K02798	1.963	0.51119	0.00012294	0.029987	4.6477	8.9458	5.5942 × 10^−15^	4.3702 × 10^−12^	ko:K02798 cmtB; mannitol PTS system EIIA component [EC:2.7.1.197]	
K03483	31.123	0.81399	0.00013157	0.029987	5.6633	8.1866	3.3345 × 10^−15^	3.9425 × 10^−12^		mtlR; mannitol operon transcriptional activator
K02810	14.768	0.38651	0.00013305	0.029987	2.0846	10.2	0.00036687	0.0029185	ko:K02810 scrA; sucrose PTS system EIIBCA or EIIBC component [EC:2.7.1.211]	
K02053	−15.036	0.39573	0.00014499	0.029987	−2.0887	7.779	0.00025198	0.0021727	ko:K02053 ABC.SP.P; putative spermidine/putrescine transport system permease protein	
K08234	16.875	0.44431	0.00014587	0.029987	2.9769	9.1374	4.4347 × 10^−9^	1.4933 × 10^−7^		yaeR; glyoxylase I family protein
K01963	12.884	0.3427	0.00017017	0.032387	2.0293	9.7298	1.3397 × 10^−5^	0.0001886	ko:K01963 accD; acetyl-CoA carboxylase carboxyl transferase subunit beta [EC:6.4.1.2 2.1.3.15]	
K03785	16.993	0.45269	0.00017412	0.032387	3.0777	9.2258	2.5298 × 10^−10^	1.0625 × 10^−8^	ko:K03785 aroD; 3-dehydroquinate dehydratase I [EC:4.2.1.10]	
K02443	−22.726	0.61306	0.0002097	0.036299	−2.1083	7.6313	0.0047712	0.020616		glpP; glycerol uptake operon antiterminator
K02054	−13.832	0.37362	0.00021374	0.036299	−1.7256	7.3454	0.0005016	0.0037598	ko:K02054 ABC.SP.P1; putative spermidine/putrescine transport system permease protein	

The base two logarithmic value of fold changes (log2FC) represents the increase (+) or decrease (−) in the abundance of a particular *Kyoto* Encyclopedia of Genes and Genomes (KEGG) *orthology* (*KO*) identifiers between the two groups. All statistical analyses showed the FDR (false discovery rate) that indicates the *p*-value after adjustment for multiple comparisons. Database accessed on 20 December 2024.

## Data Availability

The DNA sequences generated and analyzed during the current study are available in the NCBI SRA repository under BioProject PRJNA991466. https://www.ncbi.nlm.nih.gov/bioproject/?term=PRJNA991466 (accessed on 5 March 2025).

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
