# Peer review of "In Infants with Neuroblastoma Standard Therapy Only Partially Reverts the Fecal Microbiome Dysbiosis Present at Diagnosis"

_microorganisms, 2025, doi:10.3390/microorganisms13030691_

Round 1
Reviewer 1 Report
Comments and Suggestions for Authors
This peer-reviewed manuscript is devoted to the study of the gut microbiome in a very complex group of patients: infants with neuroblastoma. The authors prospectively analyzed changes in the composition, structure, and putative metabolic functions of the gut microbiota at disease onset and after standard-of-care therapy. In addition, the authors recruited a group of healthy infants with comparable physiological and biochemical parameters. Undoubtedly, this study is not only fundamental but also have a practical interest although it is not without some limitations. The manuscript describes the methods in detail and qualitatively, not only the sampling, logistics, and study design but also bioinformatics and statistical analyses. The main comments are systematized below.
1. P. 1, L. 18, 20 and further in the text: The authors used an old taxonomy of bacteria in the text of the article; therefore, it is necessary to carefully proofread and make corrections in accordance with the current taxonomy. See https://www.bacterio.net/ and Oren A, Garrity GM. Valid publication of the names of forty-two phyla of prokaryotes. Int J Syst Evol Microbiol 2021; 71:5056.
For example, Firmicutes (current name Bacillota).
Unfortunately, authors often do not consider the changes that are made in the international nomenclature, but it is necessary to use bacteriological terms carefully and accurately.
p. 12, line 313, it is necessary to check the correct and current spelling of all bacteria species: Clostridium difficile is an unused name, as its current name is Clostridioides difficile, etc.
The same type of comments should include the use of italics when writing the species and genera names of microorganisms. The authors did not provide the bacteriological nomenclature. If the authors have decided to write all bacterial taxonomy names in italics, this must be done throughout the text.
Otherwise, see p. 6, lines 214, 215; 216, 219, and so on.
2. P. 2, line 63: This manuscript does not contain any information about the control group. A table with data for the two cohorts is needed: overall infant health indicators and the statistically significant differences between them were.
3. P. 5, lines 187-200, Section "Alpha and beta diversity at onset and during treatment", as well as p. 12, lines 307-314, Section "Weighted Correlation Network Analysis": The correctness of the interpretation of the results in these sections cannot be assessed, since there are no illustrative materials to support these results.
Supplementary Figures 1 and 2 are missing.
4. Figure 1 and section "Microbiome compositional analysis of healthy infants and infants with NB": it is desirable to add statistical significance to the figure because in the text the results are discussed in this context.
The authors write “significantly” - so numbers are needed.
5. Page 6, line 251: Firmicutes is a phylum of bacteria. In this context, it is necessary to list specific genera, indicating their affiliation with the Phylum.
6. Page 7, lines 236-240: What is written here is not reflected in Figure 2D. It is necessary to align the text and its visualization.
7. Page 9, lines 281-296, Section "Sparse Correlations for Compositional data (SparCC) analysis": It is necessary to align the text and its visualization, remove repetitions, and provide correct and relevant names for the bacterial species.
For example, Cronobacter - check values - according to the figure - 50:50; Klebsiella - remove duplicate name, Morganella - no such name in the figure, Gemminger - no such bacterium, not in the figure either, if you mean Gemmiger, then these are not proteobacteria, but Bacillota; Clostridia; Eubacteriales; Eubacteriales incertae sedis. This section should be double-checked and aligned with the data visualization.
8. p. 12, line 311: the authors should explain in detail the principle by which the groups of organisms in Table 2 were formed. It was impossible to assess the accuracy of the data interpretation.
9. The authors should elucidate the strengths and weaknesses of their research in the discussion.
Author Response
Reviewer #1
COMMENT: This peer-reviewed manuscript is devoted to the study of the gut microbiome in a very complex group of patients: infants with neuroblastoma. The authors prospectively analyzed changes in the composition, structure, and putative metabolic functions of the gut microbiota at disease onset and after standard-of-care therapy. In addition, the authors recruited a group of healthy infants with comparable physiological and biochemical parameters. Undoubtedly, this study is not only fundamental but also have a practical interest although it is not without some limitations. The manuscript describes the methods in detail and qualitatively, not only the sampling, logistics, and study design but also bioinformatics and statistical analyses. The main comments are systematized below.
RESPONSE: We appreciate the Reviewer’s recognition of our work, despite the inherent limitations due to the rarity of neuroblastoma cases.
COMMENT: 1. P. 1, L. 18, 20 and further in the text: The authors used an old taxonomy of bacteria in the text of the article; therefore, it is necessary to carefully proofread and make corrections in accordance with the current taxonomy. See https://www.bacterio.net/ and Oren A, Garrity GM. Valid publication of the names of forty-two phyla of prokaryotes. Int J Syst Evol Microbiol 2021; 71:5056. For example, Firmicutes (current name Bacillota). Unfortunately, authors often do not consider the changes that are made in the international nomenclature, but it is necessary to use bacteriological terms carefully and accurately.
- 12, line 313, it is necessary to check the correct and current spelling of all bacteria species: Clostridium difficile is an unused name, as its current name is Clostridioides difficile, etc.
RESPONSE: We agree with the Reviewer that terms must be used carefully and accurately. Therefore, we have updated the bacteriological terms throughout the manuscript. In addition, we have added a reference citing the website used for the current taxonomy.
COMMENT: The same type of comments should include the use of italics when writing the species and genera names of microorganisms. The authors did not provide the bacteriological nomenclature. If the authors have decided to write all bacterial taxonomy names in italics, this must be done throughout the text.
Otherwise, see p. 6, lines 214, 215; 216, 219, and so on.
RESPONSE: We agree with the Reviewer that consistency is important. Thus, the Bacteria are now written in italics throughout the manuscript.
COMMENT 2. P. 2, line 63: This manuscript does not contain any information about the control group. A table with data for the two cohorts is needed: overall infant health indicators and the statistically significant differences between them were.
RESPONSE: As stated in the text, the control group was made of infants not affected by neuroblastoma in the same age group. Since the informed consent for unaffected infants did not ask for detailed health status, we cannot make the requested Table. However, we strongly believe that the inclusion of a control group in our study is of great value regardless of producing a detailed Table.
COMMENT 3. P. 5, lines 187-200, Section "Alpha and beta diversity at onset and during treatment", as well as p. 12, lines 307-314, Section "Weighted Correlation Network Analysis": The correctness of the interpretation of the results in these sections cannot be assessed, since there are no illustrative materials to support these results.
RESPONSE: We apologize for the missing Supplemental Figures 1 and 2, which are now included in the revised text. The values for the False Discovery Rate in the comparisons are indicated in each panel.
COMMENT Supplementary Figures 1 and 2 are missing.
RESPONSE: We apologize for the mistake. Supplementary Figures are now supplied with the revised text.
COMMENT 4. Figure 1 and section "Microbiome compositional analysis of healthy infants and infants with NB": it is desirable to add statistical significance to the figure because in the text the results are discussed in this context. The authors write “significantly” - so numbers are needed.
RESPONSE: As suggested, we have added the statistical number to the legend of Figure 1 and to each panel of Supplemental Figure 1.
COMMENT 5. Page 6, line 251: Firmicutes is a phylum of bacteria. In this context, it is necessary to list specific genera, indicating their affiliation with the Phylum.
RESPONSE: We have added the genera belonging to the phylum of Bacillota (Eubacterium, Anaerostipes, Dorea, Rosuburia, Dialister, Turicibacter).
COMMENT: 6. Page 7, lines 236-240: What is written here is not reflected in Figure 2D. It is necessary to align the text and its visualization.
RESPONSE: We apologize for the mistakes and have prepared a new Figure 2 and we modified the text accordingly on page 7, lines 23 -249.
COMMENT: 7. Page 9, lines 281-296, Section "Sparse Correlations for Compositional data (SparCC) analysis": It is necessary to align the text and its visualization, remove repetitions, and provide correct and relevant names for the bacterial species.
For example, Cronobacter - check values - according to the figure - 50:50; Klebsiella - remove duplicate name, Morganella - no such name in the figure, Gemminger - no such bacterium, not in the figure either, if you mean Gemmiger, then these are not proteobacteria, but Bacillota; Clostridia; Eubacteriales; Eubacteriales incertae sedis. This section should be double-checked and aligned with the data visualization.
RESPONSE: We modified the text on pages 11, lines 292-306.
COMMENT: 8. p. 12, line 311: the authors should explain in detail the principle by which the groups of organisms in Table 2 were formed. It was impossible to assess the accuracy of the data interpretation.
RESPONSE: Weighted Gene Co-expression Network Analysis (WGCNA) is often used to find clusters (modules) of highly correlated features in large datasets. When applied to fecal microbiomes and clinical variables, WGCNA can identify clusters of microbial taxa and clinical traits that share similar variation patterns across samples. We have added the sentences in the legend to Table 2, page 14, lines 328-334.
COMMENT: 9. The authors should elucidate the strengths and weaknesses of their research in the discussion.
As suggested by the Reviewer we have added a paragraph in the Discussion (page 17, lines 398-402) addressing the strengths and weakness of our study. Please see also the addition requested by Reviewer #2 on the benefit of our study at page 17, lines 402-413.
Reviewer 2 Report
Comments and Suggestions for Authors
microorganisms-3517196-peer-review-v1
The article is called "strait to the point". Authors have clearly decided wat is the objective and managed to perform planned experiments with target objective a provided clear evidences about differences in the microbiota in NB and control groups. This was well planned and performed work and well presented.
Maybe authors can extend a bit of an introduction with focus what specific alterations in microbiome were observed in other cancers cases in comparison with healthier individuals. Maybe some evidence of the role of microbiota as factor to prevent or stimulate cancer development will be an interesting point to be presented in the introduction.
Methods are described well and in sufficient details.
The Results and Discussion sections were presented with sufficient detail and demonstrated a deep understanding of the results obtained, offering an engaging discussion. The authors employed various research approaches to analyze the data effectively and discussed the findings appropriately.
In fact, this is a very well presented paper. However, it will be really positive if authors can build an appropriate conclusion where they can state benefits of this research and clear take home message.
References need to be formatted according to the exigences of the Publisher and Journal.
Author Response
Reviewer #2
COMMENT: The article is called "strait to the point". Authors have clearly decided wat is the objective and managed to perform planned experiments with target objective a provided clear evidences about differences in the microbiota in NB and control groups. This was well planned and performed work and well presented.
RESPONSE: We thank the Reviewer for the appreciation of our work.
COMMENT: Maybe authors can extend a bit of an introduction with focus what specific alterations in microbiome were observed in other cancers cases in comparison with healthier individuals. Maybe some evidence of the role of microbiota as factor to prevent or stimulate cancer development will be an interesting point to be presented in the introduction.
RESPONSE: As suggested by the Reviewer, we have added a paragraph in the Introduction (page 1, lines 44-49) addressing the alterations found in the gut microbiota of cancer patients in comparison with healthy individuals.
COMMENT: Methods are described well and in sufficient details.
The Results and Discussion sections were presented with sufficient detail and demonstrated a deep understanding of the results obtained, offering an engaging discussion. The authors employed various research approaches to analyze the data effectively and discussed the findings appropriately.
In fact, this is a very well presented paper.
RESPONSE: We are grateful to the Reviewer for the appreciation of our work.
COMMENT: However, it will be really positive if authors can build an appropriate conclusion where they can state benefits of this research and clear take home message.
RESPONSE: As suggested by the Reviewer we have added in the Discussion a conclusive statement on the benefit of our study (page 17, lines 402-412).
COMMENT: References need to be formatted according to the exigences of the Publisher and Journal.
RESPONSE: We have now formatted the references according to the Journal style.
Round 2
Reviewer 1 Report
Comments and Suggestions for Authors
I am satisfied with the corrections that the authors have made in the manuscript.
The only remark I would like to repeat is that Clostridium difficile currently has taxonomic status: synonym, the current name of this microorganism: Correct name: Clostridioides difficile (Hall and O'Toole 1935) Lawson et al. 2016, see https://lpsn.dsmz.de/species/clostridium-difficile.
Need to be corrected: L. 436 and 500.
Author Response
Thank you for your comments. Author has revised Clostridium difficile to Clostridioides difficile.